# Probabilistic Regular Graph Languages

## Abstract

Distributions over strings and trees can be represented by probabilistic regular languages, which characterize many models in natural language processing. Recently, several datasets have become available which represent natural language phenomena as graphs, so it is natural to ask whether there is an equivalent of probabilistic regular languages for graphs. To answer this question, we review three families of graph languages: Hyperedge Replacement Languages (HRL), which can be made probabilistic; Monadic Second Order Languages (MSOL), which support the crucial property of closure under intersection; and Regular Graph Languages (RGL; Courcelle 1991), a subfamily of both HRL and MSOL which inherits these properties, and has not been widely studied or applied to NLP. We prove that RGLs are closed under intersection and provide an efficient parsing algorithm, with runtime linear in the size of the input graph.

## 1 Introduction

NLP systems for machine translation, summarization, paraphrasing, and other problems often fail to preserve the compositional semantics of sentences and documents because they model language as bags of words, or at best syntactic trees. To preserve semantics, they must model semantics. In pursuit of this goal, several datasets have been produced which pair natural language with compositional semantic representations in the form of directed acyclic graphs (DAGs), including the Abstract Meaning Represenation Bank (AMR; Banarescu et al. 2013), the Prague Czech-English Dependency Treebank (Hajič et al., 2012), Deep-

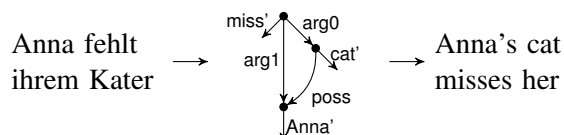

Anna fehlt ihrem Kater $\rightarrow$ [graph] $\rightarrow$ Anna's cat misses her

Figure 1: Semantic machine translation using AMR (Jones et al., 2012). The edge labels identify 'cat' as the object of the verb 'miss', 'Anna' as the subject of 'miss' and 'Anna' as the possessor of 'cat'. Edges with no head node are interpreted as node labels.

bank (Flickinger et al., 2012), and the Universal Conceptual Cognitive Annotation (Abend and Rappoport, 2013). To make use of this data, we require probabilistic models of graphs.

Consider how we might use compositional semantic representations in machine translation (Figure 1). We first parse a source sentence to its semantic representation, and then generate a target sentence from this representation. Stated more formally, we first predict a graph $G$ from a source string $s$, and then predict a target string $t$ from $G$, giving us a conditional model $\mathbb{P}(t, G|s)$, which we can decompose as $\mathbb{P}(t, G|s) = \mathbb{P}(t|G)\mathbb{P}(G|s)$. Jones et al. (2012) observe that this decomposition can be modeled with a pair of probabilistic synchronous grammars over domains of strings and graphs. Given a domain of source strings $L_s$, a domain of source graphs $L_G$, a domain of target graphs $L_{G'}$ and a domain of target strings $L_t$, these grammars define relations $R \subseteq L_s \times L_G$ and $R' \subseteq L_{G'} \times L_t$. Translation is then the solution to the following inference problem, given input $s$:

$$\arg\max_{t \in \mathcal{L}_t} \sum_{\{G|(s,G) \in R \land (G,t) \in R'\}} \mathbb{P}(t|G)\mathbb{P}(G|s)$$

In practical settings the sum over $G$ is typically replaced with an $\arg\max$. Either way, we must

define probability distributions over the graph domains and efficiently compute their intersection.

For NLP problems in which data is in the form of strings and trees, such distributions can be represented by finite automata (Mohri et al., 2008; Allauzen et al., 2014), which are closed under intersection and can be made probabilistic. It is therefore natural to ask whether there is a family of graph languages with similar properties to finite automata. Recent work in NLP has focused primarily on two families of graph languages: **hyperedge replacement languages** (HRL; Drewes et al. 1997), a context-free graph rewriting formalism that has been studied in an NLP context by several researchers (Chiang et al., 2013; Peng et al., 2015; Bauer and Rambow, 2016); and **DAG automata languages**, (Kamimura and Slutzki, 1981), studied by Quernheim and Knight (2012). Thomas (1991) showed that the latter are a subfamily of the **monadic second order languages** (MSOL), which are of special interest to us, since, when restricted to strings or trees, they exactly characterize the regular languages of each (Büchi, 1960; Büchi and Elgot, 1958; Trakhtenbrot, 1961).

The HRL and MSOL families are incomparable: that is, the context-free graph languages do not contain the regular graph languages, as is the case in languages of strings and trees (Courcelle, 1990). So, while each formalism has appealing characteristics, none appear adequate for the problem outlined above: HRLs can be made probabilistic, but they are not closed under intersection; and while DAGAL and MSOL are closed under intersection, it is unclear how to make them probabilistic (Quernheim and Knight, 2012).[1]

This paper investigates the **regular graph languages** (RGL; Courcelle 1991), defined as a restricted form of HRL (§2). The restrictions ensure that RGLs are a subfamily of MSOL. Courcelle (1991) defined regular graph grammars as an auxiliary result of a theoretical research question quite different from ours.[2] As a consequence, they have

---

[1]*Semiring-weighted* MSOLs have been defined, where weights may be in the tropical semiring (Droste and Gastin, 2005). However, for the weights to define a probability distribution, they must meet the stronger condition that the sum of multiplied weights over all definable objects is one. This does not appear to have been demonstrated for DAGAL, which violate the sufficient conditions that Booth and Thompson (1973) give for probabilistic languages. We suspect that there are DAGAL (hence MSOL) for which it is not possible.

[2]The primary research question of Courcelle (1991) is a conjecture which has only quite recently been proven (Bojanczyk and Pilipczuk, 2016).

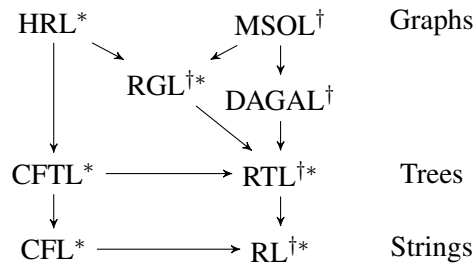

Figure 2: Containment relationships for families of regular and context-free string and tree languages, hyperedge replacement languages (HRL), monadic second order definable graph languages (MSOL), directed acyclic graph automata languages (DAGAL), and regular graph languages (RGL). ∗ indicates that the family of languages is probabilistic and † indicates that the family of languages is intersectible.

not been widely studied, and they have never been applied to NLP. We present two new results.

1. We prove that RGLs are closed under intersection (§3).

2. We give a parsing algorithm for RGL that is linear in the size of the input graph (§4).

Figure 2 summarizes the relationship of RGL to other formalisms and their properties. We conclude with a discussion of how RGL relates to other recently-discovered formalisms and its capacity to represent semantics as graphs (§5).

## 2 Regular Graph Languages

We use the following notation. If $n$ is an integer, $[n]$ denotes the set $\{1, \ldots, n\}$. If $A$ is a set, $s \in A^*$ denotes that $s$ is a sequence of arbitrary length, each element of which is in $A$. We denote by $|s|$ the length of $s$. A **ranked alphabet** is an alphabet $A$ paired with an arity function rank: $A \to \mathbb{N}$.

**Definition 1.** *A **hypergraph** over a ranked alphabet $\Gamma$ is a tuple $G = (V_G, E_G, att_G, lab_G, ext_G)$ where $V_G$ is a finite set of nodes; $E_G$ is a finite set of edges (distinct from $V_G$); $att_G : E_G \to V_G^*$ maps each edge to a sequence of nodes; $lab_G : E_G \to \Gamma$ maps each edge to a label such that $|att_G(e)| = rank(lab_G(e))$; and $ext_G$ is an ordered subset of $V_G$ called the **external nodes** of $G$.*

We assume that both the elements of $ext_G$ and the elements of $att_G(e)$ for each edge $e$ are pairwise distinct. An edge $e$ is attached to its nodes by **tentacles**, each labeled by an integer indicating the node's position in $att_G(e) = (v_1, \ldots, v_k)$. The

tentacle from $e$ to $v_i$ will have label $i$, so the tentacle labels lie in the set $[k]$ where $k = \text{rank}(e)$. To express that a node $v$ is attached to the $i$th tentacle of an edge $e$ then we say $\text{vert}(e, i) = v$. Likewise, the nodes in $\text{ext}_G$ are labeled by their position in $\text{ext}_G$. In figures, the $i$th external node will be labeled $(i)$. The **rank** of an edge $e$ is $k$ if $\text{att}(e) = (v_1, \ldots, v_k)$ (or equivalently, $\text{rank}(\text{lab}(e)) = k$). The **rank** of a hypergraph $G$ is the size of $\text{ext}_G$.

**Example 1.** Hypergraph $G$ in Figure 3 has four nodes (shown as black dots) and three hyperedges labeled $a$, $b$, and $X$ (shown boxed). The bracketed numbers $(1)$ and $(2)$ denote its external nodes and the numbers between edges and the nodes are tentacle labels. Call the top node $v_1$ and, proceeding clockwise, call the other nodes $v_2, v_3$, and $v_4$. Call its edges $e_1, e_2$ and $e_3$. Its definition would state:

$$
\begin{aligned}
att_G(e_1) &= (v_1, v_2) & lab_G(e_1) = a \\
att_G(e_2) &= (v_2, v_3) & lab_G(e_2) = b \\
att_G(e_3) &= (v_1, v_4, v_3) & lab_G(e_3) = X \\
ext_G &= (v_4, v_2).
\end{aligned}
$$

**Definition 2.** *Let $G$ be a hypergraph with an edge $e$ of rank $k$ and let $H$ be a hypergraph also of rank $k$ disjoint from $G$. The **replacement** of $e$ by $H$ is the graph $G' = G[e/H]$. Its node set $V_{G'}$ is $V_G \cup V_H$ where the $i$th node of $e$ in $G$ is fused with the $i$th external node of $V_H$. Its hyperedge set $E_{G'}$ is $(E_G - \{e\}) \cup E_H$. For each $e' \in E_{G'}$, $att_{G'}(e') = att_G(e')$ if $e' \in E_G$ and $att_{G'}(e') = att_H(e')$ if $e' \in E_H$. For each $e' \in E_{G'}$, $lab_{G'}(e') = lab_G(e')$ if $e' \in E_G$ and $lab_{G'}(e') = lab_H(e')$ if $e' \in E_H$. Its external node list is $ext'_G = ext_G$.*

**Example 2.** Replacement is shown in Figure 3. We denote the replacement as $G[X/H]$ since the edge is unambiguous given its label.

## 2.1 Hyperedge Replacement Grammars

**Definition 3.** *A **hyperedge replacement grammar** $\mathcal{G} = (N_\mathcal{G}, T_\mathcal{G}, P_\mathcal{G}, S_\mathcal{G})$ consists of a finite set of ranked nonterminal symbols $N_\mathcal{G}$, a finite set of ranked terminal symbols $T_\mathcal{G}$ (disjoint from $N_\mathcal{G}$), a finite set of productions $P_\mathcal{G}$, and a start symbol $S_\mathcal{G} \in N_\mathcal{G}$. Every production in $P_\mathcal{G}$ is of the form $X \to G$ where $X \in N_\mathcal{G}$ and $G$ is a hypergraph over $N_\mathcal{G} \cup T_\mathcal{G}$. For each production, the rank of $e$ must equal the rank of $G$.*

For each production $p : X \to G$, we will use $L(p)$ to refer to $X$ (the left-hand side of $p$) and $R(p)$ to refer to $G$ (the right-hand side of $p$). An

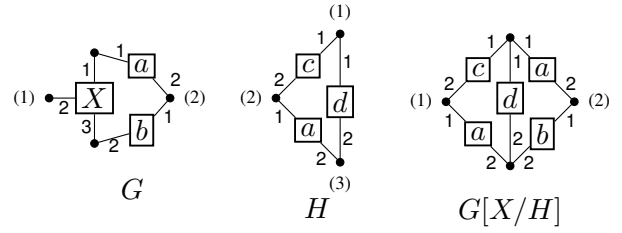

$G \qquad H \qquad G[X/H]$

Figure 3: The replacement of the $X$-labeled edge in $G$ by the graph $H$.

edge is a **terminal edge** if its label is terminal and a **nonterminal edge** if its label is nonterminal. A graph is a **terminal graph** if all of its edges are labeled with terminal symbols. The **terminal subgraph** of a graph is the subgraph consisting of all terminal edges and their endpoints.

Given a HRG $\mathcal{G}$, we say that graph $G$ **derives** graph $G'$, denoted $G \to G'$, iff there is an edge $e \in E_G$ and a nonterminal $X \in N_\mathcal{G}$ such that $\text{lab}_G(e) = X$ and $G' = G[e/H]$, where $X \to H$ is in $P_\mathcal{G}$. We extend the idea of derivation to its transitive closure $G \to^* G'$. For every $X \in N_\mathcal{G}$ we also use $X$ to denote the connected graph consisting of a single edge $e$ with $\text{lab}(e) = X$ and nodes $(v_1, \ldots, v_{\text{rank}(X)})$ such that $\text{att}_G(e) = (v_1, \ldots, v_{\text{rank}(X)})$, and we define the language $L_X(\mathcal{G})$ as $\{G \mid X \to^* G \land G \text{ is terminal}\}$. The **language of** $\mathcal{G}$ is then $L(\mathcal{G}) = L_{S_\mathcal{G}}(\mathcal{G})$. We call the family of languages that can be produced by any HRG the **hyperedge replacement languages (HRL)**.

We assume that terminal edges are always of rank 1 or 2, and depict them as directed edges where the direction is determined by the tentacle labels: the tentacle labeled 1 attaches to the source of the edge and the tentacle labeled 2 attaches to the target of the edge, if it exists.

**Example 3.** Table 1 shows a HRG deriving AMR graphs for sentences of the form 'I need to want to need to want to ... to want to go'. Figure 4 is a graph derived by the grammar. The grammar is somewhat unnatural, a point we will return to (§5).

## 2.2 Properties of HRGs

A HRG can be made probabilistic just as a CFG can: for each nonterminal symbol, we define a distribution over the right-hand sides of all productions with that symbol as its left-hand side (Booth and Thompson, 1973). HRLs are not closed under intersection, just as the CFLs (which they general-

Table 1: Productions of a HRG. The labels $p, q, r, s, t,$ and $u$ label the productions so that we can refer to them in the text. Note that $Y$ can rewrite in two ways, either via production $r$ or $s$.

Figure 4: Graph derived by grammar in Table 1.

ize) are not: emptiness of HRLs is decidable, but given two HRLs simulating CFLs, the emptiness of their intersection is undecidable.

### 2.3 Regular Graph Grammars

A regular graph grammar (RGG; Courcelle 1991) is a restricted form of HRG. To explain the restrictions, we first require some definitions.

**Definition 4.** *Given a graph $G$, a **path** in $G$ from a node $v$ to a node $v'$ is a sequence*

$$(v_0, i_1, e_1, j_1, v_1)(v_1, i_2, e_2, j_2, v_2)$$
$$\ldots (v_{k-1}, i_k, e_k, j_k, v_k) \quad (1)$$

*such that $v_0 = v, v_k = v', vert(e_r, i_r) = v_{r-1}$, and $vert(e_r, j_r) = v_r$, for each $r \in \{1, \ldots, k\}$. The length of this path is $k$.*

A path is **terminal** if every edge in the path has a terminal label. A path is **internal** if each $v_i$ is internal for $1 \le i \le k-1$. Note that the endpoints $v_0$ and $v_k$ of an internal path can be external.

**Definition 5.** *A HRG $\mathcal{G}$ is a **Regular Graph Grammar** if each nonterminal in $N_\mathcal{G}$ has rank at least one and for each $p \in P_\mathcal{G}$ the following hold:*

*(C1) $R(p)$ has at least one edge. Either it is a single terminal edge, all nodes of which are external, or each of its edges has at least one internal node.*

*(C2) Every pair of nodes in $R(p)$ is connected by a terminal and internal path.*

**Example 4.** The grammar in Table 1 is an RGG. Regular string and tree grammars are also RGGs (Figures 5 and 6).

Figure 5: RGGs generalize regular grammars.

Figure 6: RGGs generalize regular tree grammars.

We call the family of languages generated by RGGs the **regular graph languages** (RGLs).

## 3 RGLs are closed under intersection

Monadic second-order languages (MSOLs; Courcelle and Engelfriet 2011) are graph languages defined by statements in monadic second-order (MSO) logic. An MSO formula $\varphi$ defines the graph language $L(\varphi) = \{G \mid G \models \varphi\}$. A full explanation is beyond the scope of this paper but we provide a brief discussion of MSO in the supplementary materials. Courcelle (1991) proves that restrictions C1 and C2 ensure that any RGL is also a MSOL.

MSOL is trivially closed under intersection, since the conjunction of two MSO statements is

also an MSO statement: if $\varphi_1$ and $\varphi_2$ are both MSO formulae, then $L(\varphi_1) \cap L(\varphi_2) = L(\varphi_1 \wedge \varphi_2)$. However, this does not guarantee that an arbitrary subfamily of MSOL is closed under intersection—it only guarantees that the intersection of two languages in subfamily $\mathbb{G}$ are in MSOL, not necessarily in $\mathbb{G}$ itself. Here, we give a sufficient condition for a subfamily $\mathbb{G}$ to be closed under intersection.

**Proposition 1.** *Let $\mathbb{G}$ be (1) a subclass of HRG, defined as a restriction on the right-hand sides of its productions that does not depend on nonterminal labels; and (2) MSO-definable. Then for any pair of languages $L_1, L_2 \in \mathbb{G}$, the language $L_1 \cap L_2$ is also in $\mathbb{G}$.*

*Proof.* Since both $L_1$ and $L_2$ are both in HRL and MSOL, we can look at them from both perspectives. Let $\mathcal{G}_1$ be a HRG deriving $L_1$ and let $\phi_2$ be an MSO statement defining $L_2$. Propositions 1.10 and 4.8 in Courcelle (1990) prove that the intersection of a HR language and an MSO language is in HRL, by constructing a HRG which derives all and only those graphs in the intersection of the two languages.[3] This HRG has finitely many nonterminals defined by the cross product of the nonterminals of the original HRG and a finite set of 'states' of the MSO.[4] The productions of the intersection grammar are copies of the original HRG, with different nonterminal labels. Hence we can construct HRG $\mathcal{G}_\cap$ such that $L(\mathcal{G}_\cap) = L_1 \cap L_2$ and the productions in $\mathcal{G}_\cap$ satisfy any restriction that $\mathcal{G}_1$ satisfied since the restriction is in terms of the right-hand sides of the productions but not the nonterminal labels. Therefore, $\mathcal{G}_\cap$ is in $\mathbb{G}$. $\square$

RGG satisfies the conditions of Proposition 1, so RGLs are closed under intersection. Importantly, both proofs—that RGL is in MSOL and that RGL is closed under intersection—are constructive, implying that it is possible to construct the intersection grammar.

## 4 RGL Parsing

To parse RGG, we will exploit the property that every nonterminal including the start symbol has rank at least one (Definition 5), and we assume

---

[3]This is a generalization to graphs of the proof that the intersection of a context-free and regular string language is a context-free string language (Bar-Hillel et al., 1961).

[4]For each MSO-definable language $L$ in some set of all possible graphs $L'$, there exists a set $A$, a homomorphism $h : L' \to A$, and a finite subset $C$ of $A$ such that $L = h^{-1}(C)$. The finite set of 'states' here is the set $C$.

that the corresponding external node is identified in the input graph. This mild assumption may be a reasonable for applications like AMR parsing, where grammars could be designed so that the external node is always the unique root. Later we will relax this assumption.

The availability of an identifiable external node suggests a top-down algorithm, and we take inspiration from a top-down parsing algorithm the predictive top-down parsable grammars, another subclass of HRG (Drewes et al., 2015). These grammars, the graph equivalent of LL(1) string grammars, are incomparable to RGG, but the algorithms are related in their use of top-down prediction and in that they both fix an order of the edges in the right-hand side of each production.

### 4.1 Top-down Parsing for RGLs

Just as the algorithm of Chiang et al. (2013) generalizes CKY to HRG, our algorithm generalizes Earley's algorithm (Earley, 1970). Both algorithms operate by recognizing incrementally larger subgraphs of the input graph, using a succinct representation for subgraphs that depends on an arbitrarily chosen **marker node** $m$ of the input graph.

**Definition 6.** *(Chiang et al. 2013; Definition 6) Let $I$ be a subgraph of $G$. A **boundary node** of $I$ is a node which is either an endpoint of an edge in $G \backslash I$ or an external node of $G$. A **boundary edge** of $I$ is an edge in $I$ which has a boundary node as an endpoint. The **boundary representation** of $I$ is the tuple $b(I) = \langle bn(I), be(I), m \in I \rangle$ where*

*1. $bn(I)$ is the set of boundary nodes of $I$*

*2. $be(I)$ is the set of boundary edges of $I$*

*3. $(m \in I)$ is a flag indicating whether the marker node is in $I$.*

Chiang et al. (2013) prove each subgraph has a unique boundary representation, and give algorithms that use only boundary representations to compute the union of two subgraphs, requiring time linear in the number of boundary nodes; and to check disjointness of subgraphs, requiring time linear in the number of boundary edges.

For each production $p$ of the grammar, we impose a fixed order on the edges of $R(p)$, as in Drewes et al. (2015). We discuss this order in detail in §4.2. As in Earley's algorithm, we use dotted rules to represent partial recognition of productions: $X \to \bar{e}_1 \ldots \bar{e}_{i-1} \bullet \bar{e}_i \ldots \bar{e}_n$ means that we have identified the edges $\bar{e}_1$ to $\bar{e}_{i-1}$ and that we must next recognize edge $\bar{e}_i$. We write $\bar{e}$ and

$\bar{v}$ for edges and nodes in productions and $e$ and $v$ for edges and nodes in a derived graph. When the identity of the sequence is immaterial we abbreviate it as $\alpha$, for example writing $X \to \bullet \, \alpha$.

We present our parser as a deductive proof system (Shieber et al., 1995). The items of the parser are of the form

$$[b(I), p : X \to \bar{e}_1 \ldots \bullet \bar{e}_i \ldots \bar{e}_n, \phi_p]$$

where $I$ is a subgraph that has been recognised as matching $\bar{e}_1, \ldots, \bar{e}_{i-1}$; $p : X \to \bar{e}_1, \ldots, \bar{e}_n$ is a production (called $p$) in the grammar with the edges in order; and $\phi_p : E_{R(p)} \to V_G^*$ is maps the endpoints of edges in $R(p)$ to nodes in $G$.

For each production, $p$, we number the nodes in some arbitrary order. Using this, we construct the function $\phi_p^0 : E_{R(p)} \to V_{R(p)}^*$ such that for $\bar{e} \in E_{R(p)}$ if $\text{att}(\bar{e}) = (\bar{v}_1, \bar{v}_2)$ then $\phi_p^0(\bar{e}) = (\bar{v}_1, \bar{v}_2)$. As we match edges in the graph with edges in $p$, we assign the nodes $\bar{v}$ to nodes in the graph. For example, if we have an edge $\bar{e}$ in a production $p$ such that $\text{att}(\bar{e}) = (\bar{v}_1, \bar{v}_2)$ and we find an edge $e$ which matches $\bar{e}$, then we update $\phi_p$ to record this fact, written $\phi_p[\text{att}(\bar{e}) = \text{att}(e)]$. We also use $\phi_p$ to record assignments of external nodes. If we assign the $i$th external node to $v$, we write $\phi_p[\text{ext}_p(i) = v]$. We write $\phi_p^0$ to represent a mapping with no grounded nodes.

Since our algorithm makes top-down predictions based on known external nodes, our boundary representation must cover the case where a subgraph is empty except for these nodes. If at some point we know that our subgraph has external nodes $\phi(\bar{e})$, then we use the shorthand $\phi(\bar{e})$ rather than the full boundary representation $\langle \phi(\bar{e}), \emptyset, m \in \phi(\bar{e}) \rangle$.

To keep notation uniform, we use dummy nonterminal $S^* \notin N_{\mathcal{G}}$ that derives $S_{\mathcal{G}}$ via production $p_0$. For graph $G$, our system includes the axiom:

$$[\text{ext}_G, p_0 : S^* \to \bullet \, S_{\mathcal{G}}, \phi_{p_0}^0[\text{ext}_{R(p_0)} = \text{ext}_G]].$$

Our goal is to prove:

$$[b(G), S^* \to S_{\mathcal{G}} \bullet, \phi]$$

where $\phi$ is the union of the $\phi_p$s for each production $p$ used to derive $G$.

As in Earley's algorithm, we have three inference rules: PREDICT, SCAN and COMPLETE (Table 2). PREDICT is applied when the edge after the dot is nonterminal, assigning any external nodes that have been identified. SCAN is applied when the edge after the dot is terminal. Using $\phi_p$, we may already know where some of the endpoints of the edge should be, so it requires the endpoints of the scanned edge to match. COMPLETE requires that a recognized subgraph $J$ match on the external nodes of its edge in the parent graph, and that the combined subgraphs are edge-disjoint.[5]

**Example 5.** Using the RGG in Table 1, we show how to parse the graph in Figure 7, which can be derived by applying production $s$ followed by production $u$, where the external nodes of $Y$ are $(v_3, v_2)$. Assume the ordering of the edges in production $s$ is arg1, arg0, $Z$; the top node is $\bar{v}_1$; the bottom node is $\bar{v}_2$; and the node on the right is $\bar{v}_3$; and that the marker node is not in this subgraph—we elide reference to it for simplicity. The external nodes of $Y$ are determined top-down, so the parse of this subgraph is triggered by this item:

$$[(v_3, v_2), Y \to \bullet \, \text{arg1arg0}Z, \phi_s]$$

where $\phi_s(\text{arg1}) = (\bar{v}_1, v_3)$, $\phi_s(\text{arg0}) = (\bar{v}_1, v_2)$, and $\phi_s(Z) = (\bar{v}_1)$.

Table 3 shows how we can prove the item

$$[\langle \{v_3, v_2\}, \{e_3, e_2\} \rangle, Y \to \text{arg1arg0}Z \bullet, \phi]$$

The boundary representation $\langle \{v_3, v_2\}, \{e_3, e_2\} \rangle$ in this item represents the whole subgraph shown in Figure 7.

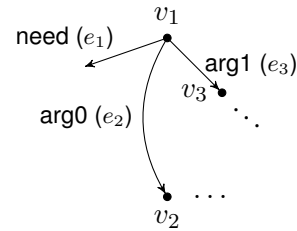

Figure 7: Top left subgraph of Figure 4. To refer to nodes and edges in the text, they are labeled $v_1, v_2, v_3, e_1, e_2$, and $e_3$.

## 4.2 Normal Ordering

Our algorithm requires a fixed ordering of the edges in the right-hand sides of each production. We will constrain this ordering to exploit the structure of RGG productions, allowing us to bound

---

[5]We provide a soundness and completeness proof of the parser in the supplementary materials.

| Name | Rule | Conditions |
|------|------|------------|
| PREDICT | $$\dfrac{[b(I), p : X \to \bar{e}_1 \ldots \bullet \bar{e}_i \ldots \bar{e}_n, \phi_p][q : Y \to \alpha]}{[\phi_p(\bar{e}_i), Y \to \bullet\, \alpha, \phi_q^0[\mathrm{ext}_{R(q)} = \phi_p(\bar{e}_i)]]}$$ | $\mathrm{lab}(\bar{e}_i) = Y$ |
| SCAN | $$\dfrac{[b(I), X \to \bar{e}_1 \ldots \bullet \bar{e}_i \ldots \bar{e}_n, \phi_p][e = \mathrm{edg}_{\mathrm{lab}(\bar{e}_i)}(v_1, \ldots, v_m)]}{[b(I \cup \{e\}), X \to \bar{e}_1 \ldots \bullet \bar{e}_{i+1} \ldots \bar{e}_n, \phi_p[\mathrm{att}(\bar{e}_i) = (v_1, \ldots, v_m)]]}$$ | $\phi_p(\bar{e}_i)(j) \in V_G \Rightarrow$ $\phi_p(\bar{e}_i)(j) = \mathrm{vert}(e, j)$ |
| COMPLETE | $$\dfrac{[b(I), p : X \to \bar{e}_1 \ldots \bullet \bar{e}_i \ldots \bar{e}_n, \phi_p][b(J), q : Y \to \alpha \bullet, \phi_q]}{[b(I \cup J), X \to \bar{e}_1 \ldots \bullet \bar{e}_{i+1} \ldots \bar{e}_n, \phi_p \cup \phi_q]}$$ | $\phi_p(\bar{e}_i) = \phi_q(\mathrm{ext}_{R(q)})$ $\mathrm{lab}(\bar{e}_i) = Y$ $E_I \cap E_J = \emptyset$ |

Table 2: The inference rules for the top-down parser.

| Current Item | Reason |
|--------------|--------|
| 1. $[(v_3, v_2), Y \to \bullet\, \mathrm{arg1arg0}Z, \phi_s]$ | Axiom |
| 2. $[\langle\{v_3, v_2, v_1\}, \{e_3\}\rangle, Y \to \mathrm{arg1} \bullet \mathrm{arg0}Z, \phi_s[\mathrm{att}(\mathrm{arg1}) = (v_1, v_3)]]$ | SCAN: 1. and $e_3 = \mathrm{edg}_{\mathrm{arg1}}(v_1, v_3)$ |
| 3. $[\langle\{v_3, v_2, v_1\}, \{e_3, e_2\}\rangle, Y \to \mathrm{arg1arg0} \bullet Z, \phi_s[\mathrm{att}(\mathrm{arg0}) = (v_1, v_2)]]$ | SCAN: 2. and $e_2 = \mathrm{edg}_{\mathrm{arg0}}(v_1, v_2)$ |
| 4. $[(v_1), Z \to \bullet\, \mathrm{need}, \phi_u^0[\mathrm{ext}_{R(u)} = \phi_s(Z)]]$ | PREDICT: 3. and $Z \to \mathrm{need}$ |
| 5. $[\langle\{v_1\}, \{e_1\}\rangle, Z \to \mathrm{need} \bullet, \phi_u[\mathrm{att}(need) = (v_1)]]$ | SCAN: 4. and $e_1 = \mathrm{edg}_{\mathrm{need}}(v_1)$ |
| 6. $[\langle\{v_3, v_2\}, \{e_3, e_2\}\rangle, Y \to \mathrm{arg1arg0}Z \bullet, \phi_s \cup \phi_u]$ | COMPLETE: 3. and 5. |

Table 3: The steps of recognising that the subgraph shown in Figure 7 is derived from productions $r_2$ and $u$ in the grammar in Table 1.

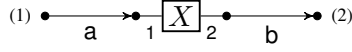

Figure 8: This graph cannot be normally ordered.

parsing complexity. If $s = \bar{e}_1 \ldots \bar{e}_n$ is an order, define $s_{i:j} = \bar{e}_i \ldots \bar{e}_j$.

**Definition 7.** *Let $s$ be the order of a right-hand side of a production. $s$ is **normal** if it has the following properties:*

*1. $\bar{e}_1$ is connected to an external node,*

*2. $s_{1:j}$ is a connected graph for all $j \in \{1, \ldots, n\}$,*

*3. if $\bar{e}_i$ is nonterminal, each endpoint of $\bar{e}_i$ must be shared with $\bar{e}_j$ where $\bar{e}_j$ is terminal and $j < i$.*

**Example 6.** The ordering of the edges of production $s$ in Example 5 is normal.

Arbitrary HRGs do not necessarily admit a normal ordering. For example, the graph in Figure 8 cannot satisfy properties 2 and 3 simultaneously. However, RGGs do admit a normal ordering.

**Proposition 2.** *If $\mathcal{G}$ is an RGG, for every $p \in P_{\mathcal{G}}$, there is a normal ordering of the edges in $R(p)$.*

*Proof.* If $R(p)$ contains a single node then it must be an external node and it must have a terminal edge attached to it since $R(p)$ must contain at least one terminal edge. If $R(p)$ contains multiple nodes then by C2 there must be terminal internal paths between all of them, so there must be a terminal edge attached to the external node, which we use to satisfy property 1. We select terminal edges once one of their endpoints is connected to an ordered edge, and nonterminal edges once all endpoints are connected to ordered edges, possible by C2. Therefore, properties 2 and 3 are satisfied. $\square$

Normal ordering tightly constrains recognition of edges. Property 3 ensures that when we apply PREDICT, the external nodes of the predicted edge are all bound to specific nodes in the graph. Properties 1 and 2 ensures that when we apply SCAN, at least one endpoint of the edge is bound.

### 4.3 Parsing Complexity

Assume a normally-ordered RGG. Let the maximum number of edges in the right-hand side of any

production be $m$; the maximum number of nodes in any right-hand side of a production $k$; the maximum degree of any node in the input graph $d$; and the number of nodes in the input graph $n$.

**Remark 1.** *The maximum number of nodes in any right-hand side of a production ($k$) is also the maximum number of boundary nodes for any subgraph in the parser.*

COMPLETE combines subgraphs $I$ and $J$ only when the entire subgraph derived from $Y$ has been recognized. Boundary nodes of $J$ are also boundary nodes of $I$ since they are nodes in the terminal subgraph of $R(p)$ where $Y$ connects. The boundary nodes of $I \cup J$ are also bounded by $k$ since are a subset of the boundary nodes of $I$.

**Remark 2.** *Given a boundary node, there are at most $(d^m)^{k-1}$ ways of identifying the remaining boundary nodes of a subgraph that is isomorphic to the terminal subgraph of the right-hand side of a production.*

The terminal subgraph of each production is connected by C2, with a maximum path length of $m$. For each edge in the path, there are at most $d$ subsequent edges. Hence for the $k - 1$ remaining boundary nodes there are $(d^m)^{k-1}$ ways of choosing them.

We count instantiations of COMPLETE for an upper bound on complexity (McAllester, 2002), using similar logic to (Chiang et al., 2013). The number of boundary nodes of $I, J$ and $I \cup J$ is at most $k$. Therefore, if we choose an arbitrary node to be some boundary node of $I \cup J$, there are at most $(d^m)^{k-1}$ ways of choosing its remaining boundary nodes. For each of these nodes, there are at most $(3^d)^k$ states of their attached boundary edges: in $I$, in $J$, or in neither. The total number of instantiations is then $\mathcal{O}(n(d^m)^{k-1}(3^d)^k)$, linear in the number of input nodes and exponential in the degree of the input graph. In the case of the AMR dataset, the maximum node degree is 17 and the average degree is 2.12.

We observe that RGGs could be relaxed to produce graphs with no external nodes by adding a dummy nonterminal $S'$ with rank 0 and a single production $S' \rightarrow S$. To adapt the parsing algorithm, we would first need to guess where the graph starts. This would add a factor of $n$ to the complexity as the graph could start at any node, requiring $n$ runs of the algorithm.

## 5 Discussion and Conclusions

RGG supports probabilistic interpretation and is closed under intersection, making it similar to regular families of string and tree languages that are widely used in NLP. The constraints of RGG also enable more efficient parsing than general HRG, and this tradeoff is reasonable since HRG is very expressive—when generating strings, it can express non-context-free languages (Engelfriet and Heyker, 1991; Bauer and Rambow, 2016), far more power than needed to express semantic graphs. On the other hand, RGG is so constrained that it may not be expressive enough: it would be more natural to derive the graph in Figure 4 from outermost to innermost predicate; but constraint C2 makes it difficult to express this, and the grammar in Table 1 does not. Perhaps we need less expressivity than HRG but more than RGG.

Since RGLs are a subfamily of both HRL and MSOL, they inherit probability and intersection closure, respectively. Courcelle (1991) discusses exactly those languages that are both HRL and MSOL, which he calls **strongly context-free languages** (SCFL).[6] SCFLs are defined non-constructively, but it is natural to ask whether other, less restrictive subfamilies of SCFLs can be constructed using similar mathematical tools. Courcelle (1991) identifies the family of series-parallel graphs as one such family, but it seems of little relevance to NLP. However, there are two recent independently-developed formalisms that may be useful. Tree-like Grammars (TLG; Matheja et al. 2015) and Restricted DAG Grammars (RDG; Björklund et al. 2016), both explicitly defined as restrictions on HRG. TLGs are in MSOL (and are closed under intersection) but we do not yet know if RDG is a subfamily of MSOL, or whether they are closed under intersection. They are both incomparable to RGG, but they share important characteristics, including the fact that the terminal subgraph of every production is connected. This means that our top-down parsing algorithm is applicable to both. In addition, if RDGs are in fact MSOL, Proposition 1 applies to them and means they are closed under intersection. We conjecture that larger, less restrictive subfamilies of SCFLs may be found based on a weaker restriction of connected terminal subgraphs, and we plan to explore these questions in future work.

---

[6]Courcelle's definition of strongly context-free is unrelated to use of this term in NLP.

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
