# Peer review of "Probabilistic Regular Graph Languages"

_ACL 2017 — decision unknown_

[Official Review · Reviewer 1 · rating 2 · confidence 5]
soundness 2 · originality 3 · clarity 2 · impact 3 · substance 3 · appropriateness 3 · meaningful comparison 3 · presentation format Poster

This one is a tough call, because I do think that there are some
important, salvageable technial results in here (notably the parsing
algorithm), but the paper as a whole has very little cohesion.        It is
united around an overarching view of formal languages in which a language
being "probabilistic" or not is treated as a formal property of the same 
variety as being closed under intersection or not.  In my opinion, what it 
means for a formal language to be probabilistic in this view has not been 
considered with sufficient rigor for this viewpoint to be compelling.

I should note, by the way, that the value of the formal results provided
mostly does not depend on the flimsiness of the overarching story.  So
what we have here is not bad research, but a badly written paper.  This needs 
more work.

I find it particulary puzzling that the organization of the paper
leaves so little space for elucidating the parsing result that
soundness and completeness are relegated to a continuation of the
paper in the form of supplementary notes.  I also find the mention of
probabilistic languages in the title of the paper to be very
disingenuous --- there is in fact no probabilistic reasoning in this
submission.

The sigificance of the intersection-closure result of section 3 is
also being somewhat overstated, I think.  Unless there is something
I'm not understanding about the restrictions on the right-hand sides
of rules (in which case, please elaborate), this is merely a matter of
folding a finite intersection into the set of non-terminal labels.

[Official Review · Reviewer 2 · rating 3 · confidence 3]
soundness 3 · originality 3 · clarity 4 · impact 3 · substance 4 · appropriateness 5 · meaningful comparison 3 · presentation format Oral Presentation

The paper is concerned in finding such a family of graph languages that is
closed under intersection and can be made probabilistic.

- Strengths:

The introduction shows relevance, the overall aim, high level context and is
nice to read.
The motivation is clear and interesting.

The paper  is extremely clear but requires close reading and much formal
background.
It nicely takes into account certain differences in terminology.

It was interesting to see how the hyper-edge grammars generalize familiar
grammars 
and Earley's algorithm.  For example, Predict applies to nonterminal edges, and
Scan applies to terminal edges.  

If the parsing vs. validation in NLP context is clarified, the paper is useful
because it is formally correct, nice contribution, instructive and can give new
ideas to other researchers.  

The described algorithm can be used in semantic parsing to rerank hypergraphs
that are produced by another parser.   In this restricted way, the method can
be part of the machinery what we in NLP use in natural language parsing and
thus relevant to the ACL.

- Weaknesses:

Reranking use is not mentioned in the introduction.

It would be a great news in NLP context if an Earley parser would run in linear
time for NLP grammars (unlike special kinds of formal language grammars). 
Unfortunately, this result involves deep assumptions about the grammar and the
kind of input. 

Linear complexity of parsing of an input graph seem right for a top-down
deterministic grammars but the paper does not recognise the fact that an input
string in NLP usually gives rise to an exponential number of graphs.  In other
words, the parsing complexity result must be interpreted in the context of
graph validation or where one wants to find out a derivation of the graph, for
example, for the purposes of graph transduction via synchronous derivations.

To me, the paper should be more clear in this as a random reader may miss the
difference between semantic parsing (from strings) and parsing of semantic
parses 
(the current work).

There does not seem to be any control of the linear order of 0-arity edges.  It
might be useful to mention that if the parser is extended to string inputs with
the aim to find the (best?) hypergraph for a given external nodes, then the
item representations of the subgraphs must also keep track of the covered
0-arity edges.                          This makes the string-parser variant
exponential.  

- Easily correctable typos or textual problems:

1)  Lines 102-106 is misleading.   While intersection and probs are true, "such
distribution" cannot refer to the discussion in the above.

2) line 173:  I think you should rather talk about validation or recognition
algorithms than parsing algorithms as "parsing" in NLP means usually completely
different thing that is much more challenging due to the lexical and structural
ambiguity.

3) lines 195-196 are unclear:  what are the elements of att_G; in what sense
they are pairwise distinct.  Compare Example 1 where ext_G and att_G(e_1) are
not disjoint sets.

4) l.206.  Move *rank* definition earlier and remove redundancy.

5) l. 267:  rather "immediately derives", perhaps.

6) 279: add "be"

7) l. 352:  give an example of a nontrivial internal path.

8) l. 472:   define a subgraph of a hypergraph

9) l. 417, l.418:  since there are two propositions, you may want to tell how
they contribute to what is quoted.

10) l. 458:  add "for"

Table:                          Axiom:              this is only place where this is
introduced as an
axiom.                    Link
to the text that says it is a trigger.

- General Discussion:

It might be useful to tell about MSOL graph languages and their yields, which
are
context-free string languages.                          

What happens if the grammar is ambiguous and not top-down deterministic? 
What if there are exponential number of parses even for the input graph due to
lexical ambiguity or some other reasons.  How would the parser behave then? 
Wouldn't the given Earley recogniser actually be strictly polynomial to m or k
?

Even a synchronous derivation of semantic graphs can miss some linguistic
phenomena where a semantic distinction is expressed by different linguistic
means.                    E.g. one language may add an affix to a verb when another
language may
express the same distinction by changing the object.  I am suggesting that
although AMR increases language independence in parses it may have such
cross-lingual
challenges.

I did not fully understand the role of the marker in subgraphs.  It was elided
later
and not really used.

l. 509-510:                 I already started to miss the remark of lines 644-647
at
this
point.

It seems that the normal order is not unique.  Can you confirm this?

It is nice that def 7, cond 1 introduces lexical anchors to predictions. 
Compare the anchors in lexicalized grammars.

l. 760.  Are you sure that non-crossing links do not occur when parsing
linearized sentences to semantic graphs?

- Significant questions to the Authors:

Linear complexity of parsing of an input graph seem right for a top-down
deterministic grammars but the paper does not recognise the fact that an input
string in NLP usually gives rise to an exponential number of graphs.  In other
words, the parsing complexity result must be interpreted in the context of
graph validation or where one wants to find out a derivation of the graph, for
example, for the purposes of graph transduction via synchronous derivations.

What would you say about parsing complexity in the case the RGG is a
non-deterministic, possibly ambiguous regular tree grammar, but one is
interested to use it to assign trees to frontier strings like a context-free
grammar?  Can one adapt the given Earley algorithm to this purpose (by guessing
internal nodes and their edges)?
Although this question might seem like a confusion, it is relevant in the NLP
context.

What prevents the RGGs to generate hypergraphs whose 0-arity edges (~words) are
then linearised?   What principle determines how they are linearised?               
  Is
the
linear order determined by the Earley paths (and normal order used in
productions) or can one consider an actual word order in strings of a natural
language? 

There is no clear connection to (non)context-free string languages or sets of
(non)projective dependency graphs used in semantic parsing.  What is written on
lines 757-758 is just misleading:  Lines 757-758 mention that HRGs can be used
to generate non-context-free languages.  Are these graph languages or string
languages?    How an NLP expert should interpret the (implicit) fact that RGGs
generate only context-free languages?  Does this mean that the graphs are
noncrossing graphs in the sense of Kuhlmann & Jonsson (2015)?